

# Random uncertainties of flux measurements by the eddy covariance technique

Üllar Rannik, Olli Peltola, Ivan Mammarella

Department of Physics, P.O. Box 48, University of Helsinki, 00014 Helsinki, Finland

*Correspondence to*: Ü. Rannik (ullar.rannik@heuristica.ee)

**Abstract.** Large variability is inherent to turbulent flux observations. We review different methods used to estimate the flux random errors. Flux errors are calculated using measured turbulent and simulated artificial records. We recommend two flux
errors with clear physical meaning: the flux error of the co-variance, defining the error of the measured flux as one standard deviation of the random uncertainty of turbulent flux observed over an averaging period of typically 30 min to 1 hour duration; and the error of the flux due to the instrumental noise. We suggest that the numerical approximation by Finkelstein and Sims (2001) is a robust and accurate method for calculation of the first error estimate. The method appeared insensitive to the integration period and the value 200 s sufficient to obtain the estimate without significant bias for variety of sites and
wide range of observation conditions. The error proposed by Wienhold et al. (1995) is a good approximation to the total flux random uncertainty provided that independent cross-covariance values far from the maximum are used in estimation as suggested in this study. For the error due to instrumental noise the method by Lenschow et al. (2000) is useful in evaluation of the respective uncertainty. The method was found to be reliable for signal-to-noise ratio, defined by the ratio of the standard deviation of the signal to that of the noise in this study, less than three. Finally, the random uncertainty of the error
estimates was determined to be in the order of 10 to 30% for the total flux error depending on the conditions and method of estimation.

## 1 Introduction

The eddy covariance (EC) method is the most direct and defensible way to measure vertical turbulent fluxes of momentum,
energy and gases between the atmosphere and biosphere. Considering an optimal measurement setup and a standardized scheme for post-field processing of the measured EC raw data, we can assume that the systematic error is minimized, and then the random error of the fluxes is typically dominating the EC flux measurement uncertainty at short time scales. The accuracy of flux random error estimates becomes important for interpretation of measurements especially when detecting small fluxes in terms of turbulent exchange or signal-to-noise ratio (SNR) of the instrumentation. Moreover, it is desirable to
estimate the total random uncertainty for each averaging period as well as to separate it into the main components, e.g. one-point sampling error and instrumental noise (Businger et al., 1996). For the uncertainty due to instrumental noise, the method proposed by Lenschow et al. (2000) have been recently applied to EC measurements not only for energy and $CO_2$ (Mauder et al., 2013; Mammarella et al., 2015), but also for $CH_4$ (Peltola et al., 2014) and $N_2O$ fluxes (Rannik et al., 2015). Few authors (Detto et al., 2011; Schmidt et al., 2012; Sturm et al., 2012; Peltola et al., 2013; Deventer et al., 2015) have used the method
proposed by Billesbach (2011) as a mean of estimating the random instrumental noise. This approach, also called "random shuffle method", consists on randomly shuffling one of the data records in time and then estimating the error as covariance between the two decorrelated time series.



Recently Langford et al. (2015) analysed in detail the uncertainties related to flux detection from the EC data with low SNR. The authors evaluated the impact of the time-lag determination and called for caution since under low SNR condition the traditional methods of maximising the cross-covariance function can lead to a systematic bias in determined fluxes. The study also reviewed the approaches for estimation of flux random errors. For quantifying the flux uncertainty Langford et al. (2015) suggest to use the method by Wienhold et al. (1995), and following Spirig et al. (2005) suggest to multiply the flux error standard deviation by a factor of three to obtain the limit of detection at 99% confidence level. The method by Lenschow et al. (2000) to calculate the effect of instrumental noise on the flux error was also validated for data with low SNR by Langford et al. (2015). They compared the method with estimates derived from the root mean square (RMS) deviation of co-variance of white noise and vertical velocity records and found that the error was not sensitive to the type of distribution of the noise and the RMS approach was consistent with the method by Lenschow et al. (2000).

In the current study we review available methods for the random error estimation of turbulent fluxes, which are widely used by the flux community. We perform calculation and analysis of flux errors by considering different error formulations described in Sect. 2.

We use the measured natural turbulent records for (i) quantitative comparison of the error estimates by Finkelstein and Sims (2001), Wienhold et al. (1995), Lenschow et al. (2000), and Billesbach (2011), and (ii) evaluation of sensitivity of error estimates on numerical approximations and calculation details. Based on the analysis we provide recommendations regarding the choice of the flux random error estimates, together with calculation guidelines for numerical evaluation.

In addition, we generate artificial records with pre-defined statistical properties characteristic to atmospheric turbulence to (iii) evaluate the flux error estimates with high accuracy. Numerically evaluated error estimates are compared with the analytical predictions to validate the theoretical expressions for different error estimates. From simulated time series the calculated error estimates allow us also to (iv) evaluate the uncertainty of the flux random errors.

## 2 Theory

Turbulent fluxes averaged over a limited time period have random errors because of the stochastic nature of turbulence (Lenschow et al. 1994; Rannik et al., 2006) as well as due to noise present in measured signals (Lenschow and Kristensen, 1985).

### 2.1 Random error of the flux

The random error of the flux defined by $F = \langle w's' \rangle = \langle (w - \langle w \rangle)(s - \langle s \rangle) \rangle$, where the angle brackets denote ensemble averaging, $w$ the vertical wind speed and $s$ the scalar, can be evaluated as the standard deviation of the co-variance, hereafter in the manuscript denoted by $\delta_F$, being the measure of one standard deviation of the random uncertainty of turbulent flux observed over an averaging period $T$. Theoretically, there are several ways to approximate the same error estimate.

For stationary time series, in the limit $T \to \infty$, the flux random error can be expressed by using the instantaneous flux $\varphi = w's' = (w - \bar{w})(s - \bar{s})$ statistics according to (Wyngaard, 1973; Lenschow et al., 1994)



$$\delta_F = \sqrt{\frac{2\tau_\varphi}{T}\left[\overline{(w's')^2} - \overline{w's'}^2\right]} \ ,$$

(1)

where the overbar denotes time averaging.

The integral time scale (ITS) of $\varphi$, $\tau_\varphi$, is defined according to

$$\tau_\varphi = \frac{1}{\sigma_\varphi^2}\int_0^\infty R_\varphi(t')dt',$$

(2)

where $R_\varphi(t') = \overline{(\varphi(t) - \overline{\varphi})(\varphi(t+t') - \overline{\varphi})}$ is the auto-covariance function of $\varphi$, $t'$ the time delay and $\sigma_\varphi^2$ is the variance

of $\varphi$. Eq. (2) can be used directly to estimate the time scale $\tau_\varphi$ by integration of the auto-covariance function of $\varphi$,

calculated from the high-frequency data records.

Rannik et al. (2009) estimated the time scale $\tau_\varphi$ and compared with simple parameterisations used in practical applications.

The time scale $\tau_\varphi$ was converted to a corresponding normalised frequency $n_\varphi$ by using $n_\varphi = \dfrac{f_\varphi(z-d)}{\overline{U}}$ and $\tau_\varphi = \dfrac{1}{2\pi f_\varphi}$.

For unstable conditions the value 0.24 (for temperature fluxes) to 0.27 (aerosol particle number concentration fluxes) was

obtained for $n_\varphi$. It was established that normalised frequency was not a function of stability under unstable conditions. Under

stable stratification the frequency $n_\varphi$ was determined to increase with stability, which was parameterised by

$$n_\varphi = 0.21\left(1 + 3.4\left(\frac{z-d}{L}\right)^{0.26}\right) \ .$$

(3)

The flux uncertainty estimate according to the spectral or the Fourier method is defined as (Lenschow and Kristensen, 1985;

Rannik and Vesala, 1999)

$$\delta_F = \left\{T^{-1}\int_{-\infty}^{\infty} S_w(f)S_s(f) + \left|S_{ws}(f)\right|^2 df\right\}^{1/2} \ ,$$

(4)

where the spectrum of time series $x = w, s$, $S_x$, can be represented as the squared magnitude of the Fourier Transform ($FT$)

of $x$, $S_x = \left|FT(x)\right|^2$, with normalisation over frequencies $f$ is assumed as $\int_{-\infty}^{\infty} S_x(f)df = \sigma_x^2$. The cross-spectrum can be

represented as $S_{ws} = FT(w)\,FT(s)^*$, where $^*$ stands for complex conjugate.

Perhaps the most frequently used method to estimate the flux error is the method equivalent to the spectral method proposed

by Lenschow and Kristensen (1985) in the time domain as

$$\delta_F = \left\{T^{-1}\int_{-T}^{T}\left[R_w(t')R_s(t') + R_{ws}(t')R_{ws}(-t')\right]dt'\right\}^{1/2}$$

(5)





in the limit $T \rightarrow \infty$, where $R_w(t') = \overline{(w(t) - \overline{w})(w(t + t') - \overline{w})}$ (and $R_s$ for scalar $s$) represents the auto-covariance and

$R_{ws}(t') = \overline{(w(t) - \overline{w})(s(t + t') - \overline{s})}$ the cross-covariance functions.

The Eq. (5) above is numerically approximated by Finkelstein and Sims (2001) as

$$\hat{\delta}_{F,FS} = \left\{ \frac{1}{n} \left[ \sum_{p=-m}^{m} \overline{w'w'}(p)\overline{s's'}(p) + \sum_{p=-m}^{m} \overline{w's'}(p)\overline{s'w'}(p) \right] \right\}^{1/2} \qquad (6)$$

with suitably chosen value of $m$, where $n$ equals the amount of data points within the averaging period and

$\overline{w'w'}(p) = \frac{1}{n-p} \sum_{i=1}^{n-p} \left( w(t_i) - \overline{w} \right)\left( w(t_{i+p}) - \overline{w} \right)$. Note that throughout of this study we use the unbiased estimates of the auto-

and cross-covariance functions.

All the error estimates presented in Sect. 2.1 are different methods for evaluation of the same flux error provided that the averaging period $T$ is much larger than the ITS $\tau_\varphi$.

## 2.2 Flux random error due to instrumental noise

Random uncertainty of the observed co-variance due to presence of noise in instrumental signal, assuming the white noise with variance independent of frequency, gives essentially the lowest limit of the flux that the system is able to measure. Such uncertainty estimate can be expressed in its simplest form as

$$\delta_{F,N} = \frac{\sigma_w \sigma_{n,f}}{\sqrt{fT}}, \qquad (7)$$

where $\sigma_w$ and $\sigma_{n,f}$ denote the standard deviation of the turbulent record of vertical wind speed and the standard deviation of instrumental noise as observed at frequency $f$. It should be noted that the signal noise at frequency $f$ can be expressed through its value at frequency 1 Hz by

$$\sigma_{n,f} = \sqrt{f}\sigma_{n,f=1Hz}, \qquad (8)$$

assuming averaging of the signal over periods $f^{-1}$ and 1 s, respectively. This enables us to re-write the Eq. (7) as

$\delta_{F,N} = \frac{\sigma_w \sigma_{n,f=1Hz}}{\sqrt{T}}$. The Eq. (7) assumes that the noise component of the vertical wind speed measurement is negligible, which is generally the case with modern sonic anemometers. The method was derived rigorously by Lenschow et al. (2000) and applied to EC fluxes by Mauder et al. (2013) to estimate the flux error due to instrumental noise. Lenschow et al. (2000) derived the method to estimate the instrumental random noise variance $\sigma_{n,f}^2$ from the auto-covariance function of the measured turbulent record close to zero-shift, enabling to determine the respective error for each half-hour flux averaging period under field conditions. In this study, the auto-covariance is linearly extrapolated to lag zero using the auto-covariance values at lags 1…5 (at 10 Hz frequency sampling rate) and the difference between this extrapolation and the observed auto-



covariance value at lag zero (i.e. the variance of the time series) is the variance related to instrumental noise. The lag interval from 0.1 to 0.5 s was chosen as a compromise between accuracy and precision of the variance estimate. This method relies on the property of the noise that it is not correlated with the true signal variation. In following, the noise variance estimate obtained according to Lenschow et al. (2000) approach is denoted by $\hat{\sigma}_{n,f}^2$ and the respective flux error according to Eq. (7)

5    by $\hat{\delta}_{F,N}$.

### 2.3 Other flux random error estimates

Wienhold et al. (1995) use a method to calculate "the error in the flux determination, the flux detection limit", calculating the standard deviation of the co-variance function $R_{ws}$ between the intervals from $-50 \le \Delta t_N \le -40$ and $40 \le \Delta t_N \le 50$ as

$$\hat{\delta}_{F,W} = \left\{ \frac{1}{2p} \sum_{i=P_1}^{P_1+p} \left[ \left( R_{ws}(i\,\Delta t) - \overline{R}_{ws} \right)^2 + \left( R_{ws}(-i\,\Delta t) - \overline{R}_{ws} \right)^2 \right] \right\}^{1/2} , \tag{9}$$

where $\Delta t_N = \frac{\Delta t \overline{U}}{Z}$ is the normalised interval of the record, $P_1 = \left\| 40 \frac{Z}{\overline{U}} \right\|$ and $p = \left\| 10 \frac{Z}{\overline{U}} \right\|$, $\overline{U}$ the average wind speed and $Z=z-d$ the measurement height relative to displacement height $d$, and $\overline{R}_{ws} = \frac{1}{2p} \sum_{i=P_1}^{P_1+p} \left[ R_{ws}(i\,\Delta t) + R_{ws}(-i\,\Delta t) \right]$. The method calculates several values of the co-variance at lags where correlation between the time series $w$ and $s$ has vanished, and assuming these are all independent estimates, calculates the standard deviation as the error estimate. Provided that the

15    variance of the co-variance function is calculated from the independent values at long enough lags, this method is equivalent to calculating the error estimate according to

$$\delta_{F,W} = \left\{ T^{-1} \int_{-T}^{T} R_w(t') R_s(t') \, dt' \right\}^{1/2} , \tag{10}$$

which is expected to underestimate the total flux error presented by Eq. (5). On the other hand it produces larger error values compared to the flux error due to instrumental noise as defined by Eq. (7) and compared to the error estimate by Billesbach

20    (2011, Eq. (11) below). We also demonstrate below (Sect. 4.1) that the subsequent values of $R_{ws}$ are not statistically fully independent and the numerical estimate Eq. (9) converges to the theoretical expression Eq. (10) only if independent values of $R_{ws}$ are selected for calculation of the standard deviation as the error estimate $\hat{\delta}_{F,W}$.

Billesbach (2011) proposed a method to calculate the flux error estimate, which according to the authors was "designed to only be sensitive to random instrument noise". The error is calculated according to

$$\hat{\delta}_{F,B} = \frac{1}{n} \sum_{i=1}^{n} w'(t_i) s'(t_j), \tag{11}$$

where $j \in [1..n]$ but the values are in the random order. Random shuffling of the time series $s$ with respect to $w$ essentially decorrelates $s$ (assumed to consist of the sum of turbulent signal and instrumental noise) from $w$, resulting in two independent variables. This error estimate is equivalent to Eq. (7) as modified to





$$\delta_{F,B} = \frac{\sigma_w \sigma_s}{\sqrt{fT}},$$ (12)

where the scalar time series variance is the sum of the variances of turbulent scalar concentration and that of the noise, i.e. $\sigma_s^2 = \sigma_c^2 + \sigma_{n,f}^2$. Therefore the method interprets the variance of turbulent variation as a part of the random noise and produces an error estimate that overestimates the flux error due to instrumental noise only. Also, since turbulence spectrum does not follow the property described by Eq. (8), the error estimate according to Eq. (11) becomes dependent on the choice of the frequency $f$ and cannot therefore be considered as a robust method to estimate the flux error.

Lenschow and Kristensen (1985) have shown that the auto-covariance function for the Poisson type of noise has the form

$$R_s(t) = \sigma_s^2 \begin{cases} 1 - \dfrac{|t|}{\Delta t}, & |t| \leq \Delta t \\ 0, & |t| \geq \Delta t \end{cases}$$ with $\Delta t = f^{-1}$. Following the formal presentation in Eq. (5) and considering that $w$ and $s$

are independent due to shuffling, making the term corresponding to the product of the cross-covariances vanish, the error estimate for the method by Billesbach (2011) becomes

$$\delta_{F,B} = \sigma_s \left\{ T^{-1} \int_{-\Delta t}^{\Delta t} \left[ R_w(t') \left( 1 - \frac{|t'|}{\Delta t} \right) \right] dt' \right\}^{1/2},$$ (13)

which after integration becomes equivalent to Eq. (12).

**2.4 The random error of the ensemble average flux**

If an average over fluxes $F_i$ ($i = 1..N$) is calculated, each of these representing a flux value observed over averaging period $T$ and being characterised by an error $\delta_{F,i}$, then the error of the average flux $\langle F \rangle = \frac{1}{N} \sum_{i=1}^{N} F_i$ can be expressed as

$$\Delta_{<F>} = \sqrt{\frac{\sum_{i=1}^{N} (\delta_{F,i})^2}{N^2}}.$$ (14)

**3 Materials and methods**

**3.1 Sites and measurements**

Measurements from three different and contrasting sites with different surface properties and observation heights of about 23 m (forest site in Hyytiälä, SMEAR II), 2.7 m (Siikaneva fen site) and 1.5 m (Lake Kuivajärvi) above surface were used to evaluate the flux error estimates for June 2012 (July 2012 at Kuivajärvi) for measured temperature, carbon dioxide ($CO_2$), water vapour ($H_2O$) and methane ($CH_4$, Siikaneva only) fluxes.



### 3.1.1 SMEAR II (Forest site)

The first set of measurements was done at the SMEAR II station (Station for Measuring Forest Ecosystem-Atmosphere Relationships), Hyytiälä, Southern Finland (61°51′ N, 24°17′ E, 181 m ASL). The station is surrounded by extended areas of coniferous forests and the tower of the EC measurements is located in a 50-years-old (in 2012) Scots pine (*Pinus sylvestris L.*) forest with dominant tree height of 17 m. The EC measurements were performed at 23 m height, approximately 6 m above the forest canopy. The wind speed components and sonic temperature were measured by an ultrasonic anemometer (Solent Research 1012R2, Gill Ltd, UK), and fast response $CO_2$ and $H_2O$ mole fraction by an infrared gas analyser (LI-6262, LI-COR Inc., Lincoln, NE, USA). Description of the measurement in micrometeorological context at SMEAR II station can be found in Rannik (1998), more detailed description of the station and the measurements in Hari and Kulmala (2005).

### 3.1.2 Kuivajärvi (Lake site)

The second dataset is taken from Lake Kuivajärvi (61°50′N, 24°17′E), located close to the Hyytiälä Forestry Field Station and SMEAR II Station. Lake Kuivajärvi is a small humic boreal lake extending about 2.6 km in northwest to southeast direction, and it is a few hundred meters wide (surface area is 0.63 km$^2$). The measurement platform, firmly anchored from all the four corners, is located approximately 1.8 km and 0.8 km from the northern end and southern end, respectively. Turbulent fluxes of momentum, heat, $CO_2$, and $H_2O$ are measured by an EC system (located on the above mentioned platform), which includes an ultrasonic anemometer (Metek USA-1, GmbH, Elmshorn, Germany) to measure the three wind velocity components and sonic temperature and the enclosed path infrared gas analyzer LI-7200 (LI-COR Inc., Lincoln, NE, USA) that measures $CO_2$ and $H_2O$ concentrations. The data are sampled at 10 Hz, and the gas inlet is at 1.7 m above the water surface close to the sonic anemometer. More details about the site and measurements can be found in Mammarella et al. (2015).

### 3.1.3 Siikaneva (Fen site)

The third dataset was collected at Siikaneva fen site (61˚49.961' N, 24˚11.567' E). The EC data used in this study were measured with a 3D sonic anemometer (Metek USA-1, GmbH, Elmshorn, Germany) and one closed-path analyser (LI-7000, LI-COR Inc., Lincoln, NE, USA) for $CO_2$ and $H_2O$. The sonic anemometer and the gas inlet was situated at 2.75 m above peat surface and the air was drawn to the analyser through a 16.8 m long heated sampling line. $CH_4$ mole fraction was measured with a closed-path gas analyser (FMA, Los Gatos research, USA). Further details about the site and measurements can be found from Peltola et al. (2013).

### 3.2 Flux processing

Turbulent fluxes and other statistics reported in the study were calculated over 30 min averaging period by block averaging approach (i.e. no detrending was applied if not mentioned otherwise) using the EddyUH software (Mammarella et al., 2016).





Prior to flux calculation, raw data despiking, conversion of $CO_2$ and $H_2O$ from wet to dry mole fraction and two-dimensional co-ordinate rotation of wind vector was performed (Kaimal and Finnigan, 1994). The fluxes were corrected for frequency response underestimation at low and high frequencies by using the co-spectral transfer functions calculated according to Rannik et al. (1999) and Mammarella et al. (2009) respectively, together with in-situ parameterisation of the co-spectral

model derived from ensemble mean of measured temperature co-spectra (Mammarella et al., 2009; Peltola et al., 2013; Mammarella et al., 2015).

The measured data (wind speed and concentration records) were quality screened for spikes (all 30 min periods with a single data point exceeding physically meaningful value excluded), and according to Vickers and Mahrt (1997) by applying the following statistics and selection thresholds: data with concentration skewness outside (-2, 2), or kurtosis outside (1, 8), or

Haar mean and Haar variance exceeding 2 were rejected. In addition, flux data were rejected if the 2nd coordinate rotation angle was outside the range (-15°,15°) and at Kuivajärvi data were rejected if the wind was not blowing along the lake (directions 345°–135° and 170°–290°). Additional quality screening was performed for flux stationarity by using the threshold value 1 (Foken and Wichura, 1996).

### 3.3 Superimposing Gaussian noise to the measured records

SNR is defined in the current study as $SNR = \dfrac{\sqrt{\sigma_s^{\,2} - \sigma_{n,f}^2}}{\sigma_{n,f}}$. Records with low noise level (sonic temperature at the three

sites) were used to evaluate the performance of the Lenschow et al. (2000) and Billesbach (2011) methods (Sect. 4.3.1 and 4.3.2, respectively), superimposing the measured signal with Gaussian noise. Natural variability of records in combination with different noise levels $\sigma_{n,f=10Hz}$ (0.025, 0.152 and 0.30 K) led to a range of SNR-s from about 0.3 to 20, enabling to

determine the range and threshold of SNR where the Lenschow et al. (2000) method can be used. In addition, the temperature signal was high pass filtered with a simple first-order low-pass filter in order to simulate scalar measurements ($CO_2$, $H_2O$, $CH_4$, etc.) with a closed-path analyser performing as the low-pass filter to measured signal. Low-pass filtering was executed using 0.1 s, 0.3 s and 0.6 s time constants. However, since the results for different time constants did not differ qualitatively, we present only the results for the time constant 0.3 s (Sect. 4.3.1).

### 3.4 Simulation of artificial records

We generated artificial records with pre-defined statistical properties characteristic to atmospheric turbulence. Gaussian probability density functions were assumed for vertical wind speed and concentration time series. The atmospheric surface layer similarity relationships were assumed for the variances of the records and the time scales of the auto-correlated

processes were defined via normalised frequencies (Appendix A).

The analysis was carried out as following. First, we calculated the flux errors according to analytical expressions (A4), (A5) and (12) for the three flux errors $\delta_F$, $\delta_{F,W}$, and $\delta_{F,B}$, respectively.





Second, we calculated from the repeated simulated artificial records time series (N = 10 000) the fluxes and evaluated the error estimates according to

$$\delta_F = \sigma_F = \sqrt{\langle F^2 \rangle - \langle F \rangle^2}, \tag{15}$$

where $\langle\ \rangle$ denotes ensemble averaging over a large number of records with duration $T = 30$ min. In order to estimate $\delta_{F,W}$, we retained auto-correlation of the series $w$ and $c$ but assumed no cross-correlation, i.e. with $\alpha_{wc} = 0$. In order to obtain the estimate for $\delta_{F,B}$ we assumed uncorrelated time series $w$ and $c$ by taking $\alpha_w = 0$, $\alpha_c = 0$ and $\alpha_{wc} = 0$ in Eqs. (A1) and (A2).

Third, we evaluated the errors estimates $\hat{\delta}_{F,FS}$, $\hat{\delta}_{F,W}$ and $\hat{\delta}_{F,B}$ according to Eqs. (6), (9) and (11), respectively. These error values allow us to evaluate also the uncertainty of the random flux errors via calculation of the variance of the error estimates

$$\sigma(\hat{\delta}_F) = \sigma_{\hat{\delta}_F} = \sqrt{\langle \hat{\delta}_F^2 \rangle - \langle \hat{\delta}_F \rangle^2}. \tag{16}$$

## 4 Results

### 4.1 Evaluation of error estimates based on simulated time series

The random errors of the co-variance time series as presented in Appendix A can be derived analytically as the total error (Eq. A4) and as the error for the Wienhold et al. (1995) method (Eq. A5). Assuming the atmospheric surface layer similarity forms, the relative flux errors defined by $\delta_F |F|^{-1}$ vary with wind speed and stability (Fig. 1). The error $\delta_{F,W}$ ignores the covariance part of the error expression and is therefore slightly smaller. Nevertheless, the method by Wienhold et al. (1995) provides good approximation of the total flux error $\delta_F$. The error according to the Billesbach (2011) method ($\delta_{F,B}$) is much lower (estimated according to Eq. 12) and the relative error does not show dependence on wind speed.

According to ASL similarity functions the relative flux error is largest at near-neutral stability and decreases both for unstable and stable conditions. In stable case the errors are smaller because turbulent spectrum is shifted towards higher frequencies resulting in more efficient averaging (over the same period $T$) and reduced relative random uncertainty. This is similar to the effect of wind speed where higher wind speed implies higher frequency turbulence and lower relative random uncertainty. In unstable case the normalised frequencies $n_w$ and $n_c$ are independent of stability and the stability dependence of the relative error is caused by the functions in Eq. (A6).

Further we analysed in detail the flux errors for conditions characterised by $Z\,U^{-1} = 10$ s and three stability cases $Z\,L^{-1} = $ -1, 0, 1. Table 1 indicates that the simulated error estimates according to Eq. (15) are close to the analytical expectations. Also the numerical methods by Finkelstein and Sims (2001) and Billesbach (2011) produce similar values. However, the error estimated according to the Wienhold et al. (1995) method, $\hat{\delta}_{F,W}$ in Eq. (9), deviates from what we expected theoretically (Eqs. 10 and A5). The underestimation is particularly evident under neutral conditions when the relative error is largest. We



modified the method to essentially decorrelate the individual $R_{ws}$ values by using values after every 10 second time shift and obtained much better correspondence to the expectations according to theory and simulations. Therefore we believe the error estimate $\hat{\delta}_{F,W}$ can underestimate the flux error not only because it omits the co-variance term but also because the subsequent values of $R_{ws}$ are not independent, leading to underestimation of the total variability.

The variability of the error estimates (calculated according to Eq. 16) is around 10 to 20% for $\hat{\delta}_{F,FS}$ and is slightly larger for $\hat{\delta}_{F,W}$. The uncertainty of the random error estimate is larger for unstable and neutral conditions and smaller for stable.

**4.2 Comparison of flux errors**

    The flux uncertainty estimates increase approximately linearly with the flux magnitude at all sites (Fig. 2 and Table 2). As

expected from theory, the error estimate $\hat{\delta}_{F,FS}$ yielded largest values for the uncertainty. The error $\hat{\delta}_{F,W}$ gave few tens of percentages smaller uncertainty estimates ($CO_2$: Forest 29%, Fen 35%, Lake: 45%; $H_2O$: Forest 33 %, Fen 23%, Lake 15%; $CH_4$: Fen 18 %) than $\hat{\delta}_{F,FS}$. This is related to the difference between Eqs. (5) and (10): the estimate $\hat{\delta}_{F,FS}$ included a cross-covariance term, whereas $\hat{\delta}_{F,W}$ estimated the flux uncertainty related only to the auto-covariance term. Also, the method by Wienhold et al. (1995) as defined by Eq. (9) likely underestimates the error as discussed in Sect. 4.1. The error $\hat{\delta}_{F,B}$

obtained according to Lenschow et al. (2000) gave systematically higher uncertainty estimates than $\hat{\delta}_{F,N}$, as predicted in Sect. 2.3. As $\hat{\delta}_{F,N}$ is expected to estimate the flux uncertainties due to instrumental noise, then $\hat{\delta}_{F,B}$ is clearly a different error estimate.

    Based on the linear regression statistics presented in Table 2, a few findings can be emphasised. The intercept values are small (compared to the flux error magnitudes) and imply that flux uncertainties tend to vanish with no turbulent exchange.

Generally the relative flux error is larger over the Forest (slope 0.14 and 0.18 for $CO_2$ and LE) compared to Fen site (respective slopes 0.08 and 0.09). Surprisingly, the relative flux error is largest for $CO_2$ over the Lake. This could be due to advective conditions for $CO_2$. During the calm conditions, in particular at nights, $CO_2$ is expected to drain downhill towards the lake and accumulate, causing the concentration to increase and induce variation which is not related to local exchange over the Lake. Additional variance in concentration record would impact also the flux error estimate.

The error due to instrumental noise ($\hat{\delta}_{F,N}$) is weakly correlated with flux value as expected from theory. The method $\hat{\delta}_{F,B}$ gives correlated estimates with fluxes and this is expected due to correlation between the concentration variance and the flux (turbulent exchange naturally gives rise to concentration fluctuations).

    For qualitative comparison with the behaviour of our theoretical model based on ASL similarity theory we constructed a plot similar to the one presented in Sect. 4.1 (Fig. 1), see Fig. 3. The figure illustrates that the observed behaviour holds: the



relative flux errors increase with increasing $(z-d)\,U^{-1}$ (i.e. with lower wind speeds) and the stability dependence looks similar. Our theoretical model predicted the highest relative errors for near-neutral conditions. This holds for $CO_2$ but in case of $H_2O$ the peak is shifted towards stable stratifications side. This apparent dissimilarity of scalars could be the result of different source-sink behaviour.

The method $\hat{\delta}_{F,B}$ produces wind speed invariant relative error estimates and again similar behaviour with stability as presented in Fig. 1.

In most cases the flux random uncertainty is dominated by the stochastic nature of turbulence and the instrumental noise is a minor part of the total uncertainty. The uncertainty estimate calculated by the Finkelstein and Sims (2001) method accounts for the total uncertainty of the covariance and thus the relative contribution of instrumental noise to the total uncertainty can

be assessed by comparing the flux error estimates $\hat{\delta}_{F,FS}$ and $\hat{\delta}_{F,N}$. Fig. 4 shows that for $CO_2$ flux measurements at the Fen site the instrumental noise causes around 5…10 % of the flux random uncertainty indicating that instrumental noise level is low enough for flux measurements at the given site. For sites with very low fluxes the situation might be opposite and the instrumental noise becomes limiting in detection of surface exchange.

**4.3 Numerical guidelines for error calculation**

**4.3.1 Instrumental noise according to Lenschow et al. (2000)**

The calculated $\hat{\sigma}_{n,f}^2$ values were grouped according to SNR and ITS $\tau_\varphi$. Note that the ITS for $\varphi$ is smaller roughly by a factor of 3 compared to the time scale of concentration $c$ (cf. Eqs. 3 and A8). The results are shown in Fig. 5. At all three sites, the method by Lenschow et al. (2000) overestimated the noise variance if SNR>3, the ITS was small and temperature

signal was not low-pass filtered (cf. Fig. 5a…5c). In these cases the signal was high compared to noise (SNR>3), large part of the signal was at high frequencies (implying small ITS) and the high frequencies were not attenuated. When the ITS was larger, the noise was estimated more accurately, especially at the Lake and Fen sites. On the other hand, if SNR<3, the accuracy of the noise estimation did not significantly depend on the ITS, and the relative error of the noise estimation was in general within 10 %.

If the temperature signal was low-pass filtered before superimposing the signal with Gaussian noise, the accuracy of the noise estimation was improved (cf. Fig. 5d…5f). When the SNR was below 5, the noise variance was estimated successfully (relative error within ±30 %), regardless of the ITS. Increasing the ITS improved the results, especially at the fen site (cf. Fig 5e). In addition, stronger high frequency signal damping generally increased the accuracy of the noise estimate (not shown). This result suggests that the Lenschow et al. (2000) method is more suitable for signals which are high-pass filtered

(measurements with closed-path gas analysers) than for less attenuated signals (measurements with open-path gas analysers and sonic anemometers).

On the whole, the accuracy of the Lenschow et al. (2000) method depends on how strong is the signal relative to noise at high frequencies, since the noise is estimated using small time shifts close to the auto-covariance peak. The signal-to-noise ratio at high frequencies decreases if i) the total SNR decreases, ii) the ITS increases (power spectrum shifts to lower





frequencies) or iii) the high frequency variation in the signal is dampened. Thus for instance for signals measured at a tall tower with a closed-path analyser, the method should work well in estimating the instrumental noise, since most of the turbulent signal is at relatively low frequencies (high measurement height) and the high frequency variation in the signal is dampened. On the other hand, for measurements close to the ground with an open-path analyser the method does not

perform equally well under all conditions. The reliability of the Lenschow et al. (2000) in estimating the instrumental noise from the signal is determined by the SNR, as illustrated in the Fig. 5. Therefore, a priori knowledge on the instrumental precision characteristics is needed when analysing the outcome of the method.

We analysed in more detail the performance of the Lenschow et al. (2000) method in Fig. 6. In case of low SNR and high ITS the method estimates the true noise with relatively small bias (Fig. 6 a, d). For the same SNR but low ITS value the

variance is strongly over estimated (Fig. 6 b, e). We argued earlier that low-pass filtered signals enable to obtain better noise variance estimates (Fig. 5). However, in case of high SNR (Fig. 6 c, f) the method leads to significant under estimation of the true variance. Thus filtered signals (instruments with not perfect frequency response) are not always preferred in terms of the method's ability to determine the signal noise.

In Table 3 we report the estimated signal noise statistics for the instruments used in the current study by defining reliable

values according to criterion SNR<3. The fraction of reliable estimates is low for the instruments with high precision characteristics. For example, the method does not typically work for estimation of the precision of wind speed measurements of the sonic anemometers. The estimated signal noise characteristics are in good correspondence with instrument specifications (where available) except for the $CH_4$ analyser, which had much better precision value than reported by the manufacturer.

### 4.3.2 Random error according to Billesbach (2011)

Billesbach (2011) introduced the so-called "random shuffle" method to estimate the instrumental noise from EC measurements. However, as argued in Sect. 2.3 this method does not estimate the flux error due to the instrumental noise since it mixes turbulent variation with noise and thus the error corresponding to the sum of the variances of the turbulent

signal and the noise is deduced by the method. This is exemplified by Fig. 7a and 7b: the error estimated with the "random shuffle" method ($\hat{\delta}_{F,B}$) is equal to the calculation using the combined variances of signal and noise (Fig. 7b) and not to the calculation using the variance of the noise only (Fig 7a). Thus the method cannot be expected to be suitable for estimation of the flux error due to (instrumental) noise in the signal.

### 4.3.3 Integration time in the Finkelstein and Sims (2001) method

Under different wind speed and stability conditions the ITS of turbulence varies and therefore it becomes relevant what would be the appropriate integration time for the method by Finkelstein and Sims (2001). The integration time of the flux error $\hat{\delta}_{F,FS}$ is studied by varying $m$ in Eq. (6) up to 1500 s. Further, to see the influence of different high-pass filtering techniques used in EC flux calculation, we applied the method to time series of sonic temperature from the Forest site with

following detrending options: mean removal (no detrending applied), linear detrending and auto-regressive high pass



filtering with time constant 200 s. In general, the flux error estimate increases with integration time up to about 300 s (Fig. 8). At short integration times the high-pass filtered time series converge faster to the limit at longer integration times. This indicates contribution of low-frequency part of the spectra to the error estimates. Possibly part of this low frequency variation is contributed by non-stationarity of the series. At larger integration times than about 300 s the error estimates

essentially do not change. We choose further to normalise the error estimates with average value over the interval from 400 to 600 s.

Plots for different sites (varying the observation level and surface type), and wind speed and stability influences as reflected by ITS classes indicate that integration time 200 s serves as an optimal choice for all conditions (Fig. 9). This would guarantee less than 10% systematic underestimation of the flux error even in case of 25% largest ITS values (ITS > 75[th]

percentile) for Fen and Lake sites. The figure also illustrates that the cross-covariance term in Eq. (5) contributes 10 to 30% of the error estimate, suggesting that the method by Wienhold et al. (1995), which ignores this term, underestimates the error by the same fraction.

## 5 Discussion and conclusions

Commonly applied random error estimates of turbulent fluxes were tested and compared in this study. The method proposed by Finkelstein and Sims (2001), the error estimate $\hat{\delta}_{F,FS}$ according to Eq. (6), approximates the random flux error defined as one standard deviation of the random uncertainty of turbulent flux observed over an averaging period $T$. Wienhold et al. (1995) defined an error estimate ($\hat{\delta}_{F,W}$, Eq. (9) in this study), calculating the standard deviation of cross-covariance function over the lag interval far from the maximum. They called the error estimate as the detection limit of the flux. It was shown in

the current study that the error estimate $\hat{\delta}_{F,W}$ is in a good correspondence with $\hat{\delta}_{F,FS}$ even though it does not rigorously define the same flux error. The method $\hat{\delta}_{F,W}$ underestimates the flux random uncertainty by a few tens of percent owing to the fact that it ignores the co-variance part of the estimate in Eq. (5). We also demonstrated in this study that the error estimate $\hat{\delta}_{F,W}$ as formulated by Wienhold et al. (1995) underestimates the true flux uncertainty due to the fact that the cross-covariance estimates at neighbouring lags are not independent. To overcome this deficiency we suggest to calculate the flux

error variance from the cross-correlation values over longer lag interval but separated in time. For example, in the numerical exercise we chose the cross-covariance values with 10 s intervals within the lag ranges from -300 to -100 and +100 to +300 s. The modified approach reproduced the flux error values close to theoretical expectations whereas the original method underestimated the theoretical value up to 26% (from three studied cases).

An alternative to one point statistical estimation of the flux random errors as described in this study (Sect 2.1) is the two

tower approach, where the flux random error is evaluated by using the difference of the fluxes measured at two EC towers (e.g., Hollinger et al., 2004). The method assumes statistically similar observation conditions with independent realisations of turbulence at the two towers. Since the conditions are difficult to realise because of spatial correlation in measurements (e.g. Rannik et al., 2006), we suggest that the one-point statistical approach provides rigorous but more convenient method to estimate the flux random errors. Nevertheless, the two tower approach was shown to give close results to the method by



Finkelstein and Sims (2001) when similar weather conditions at the two sites were included in the analysis (Post et al., 2015).

The error estimates with very clear physical meaning are the total error resulting from stochastic nature of turbulence due to limited sampling in time and/or in space, the method by Finkelstein and Sims (2001) and to a good approximation the

method by Wienhold et al. (1995), and the random error due to instrumental noise only. To estimate the latter from the field measurements (not from laboratory experiment) Lenschow et al. (2000) suggested to calculate the signal noise variance from the difference between the signal auto-covariance at zero lag and the extrapolated value of the auto-covariance function to zero lag. The noise variance enables to calculate the flux error according to Eq. (7), which gives essentially the flux uncertainty under conditions of no turbulent exchange (and thus variability) of scalar concentration.

Billesbach (2011) proposed the flux error estimate based on the product of vertical wind speed and concentration fluctuations, randomly re-distributing one of the series (denoted by $\hat{\delta}_{F,B}$, Eq. (11) in this study). The method was called as the "shuffling method" and the authors proposed that the method was designed to only be sensitive to random instrument noise. We point out in this study that the method effectively adds the variance of turbulent scalar variation to noise variance and therefore the method is not equivalent to (overestimating) the method proposed by Lenschow et al. (2000) and also not

to the Finkelstein and Sims (2001) method by strongly under estimating the total flux uncertainty.

Different flux error estimates have been assigned the meaning of the flux detection limit. For example, Wienhold et al. (1995) called their method as "detection limit of the flux". Billesbach (2011) suggested that the method they introduced "was sensitive to random instrument noise". The method by Lenschow et al. (2000) estimates the flux value that the system is able to detect within an averaging period $T$ under hypothetical conditions of no turbulent variation of concentration. This error

estimate serves as the theoretical lowest detection limit of the EC system. However, under natural turbulent exchange conditions the flux random uncertainty is contributed in addition to signal noise also by the stochastic nature of turbulence and the total flux error is larger, also meaning that the detection limit is larger than compared to the error introduced by the instrumental noise. Respectively, Langford et al. (2015) have defined "limit of detection" as $3x\,\hat{\delta}_{F,W}$ (the uncertainty according to Wienhold et al. (1995) method) to give the flux measurement precision within 99% confidence interval. By

default most of the publications refer to one standard deviation of flux random variability (which corresponds to 68% confidence intervals assuming normal distribution) when talking about the flux precision or random errors. If different confidence level is aimed, as by Langford et al. (2015), this should be explicitly stated.

The flux detection limit has been used also in conjunction with other flux measurement techniques. For example, in case of chamber measurements the flux detection limit has been used to denote the flux error arising from all possible error sources.

The traditional way to perform chamber measurements is to determine the gas concentration at several time moments during the chamber operation. In such data collection the sources of uncertainty are the imprecision related to gas sampling (either manual or automatic) as well as instrumental uncertainty (e.g. Venterea et al., 2009), leading to a measurement precision which is called a detection limit of chamber based flux measurement system. It has to be noted that the flux detection limit of the chamber systems depends on several factors such as the type of the chamber and respective sampling method, the





precision of the instrument, chamber dimensions and operation time. Therefore the flux detection limit of the chamber based systems (which accounts for all possible sources of uncertainty) is comparable to the total stochastic error of the EC fluxes.

We also studied the performance of the Lenschow et al. (2000) and Finkelstein and Sims (2001) flux error estimation methods over different ecosystems and observation conditions. The performance of the Lenschow et al. (2000) method is

affected by the SNR and the ITS of turbulence. We established that the method provides reliable estimates for SNR < 3 (in statistical sense, single biased values can occur). However, no criterion based on the ITS could be provided as the results deviated among sites.

Application of the EC method requires stationarity of time series within averaging period (e.g. Foken and Wichura, 1996). Non-stationarity results in higher random uncertainty of the flux value and therefore the stationarity requirement has to be

fulfilled if each 30 min or 1 hour average value is expected to be statistically significant. We tested sensitivity of the flux errors derived by the Finkelstein and Sims (2001) method on integration time and high-pass filtering of the fluxes performed by mean removal, linear detrending and auto-regressive filtering. It was observed that the flux error increased with integration time up to about 300 s revealing the influence of the low-frequency (possibly non-stationary) signal variance on the flux estimates. The high-pass filtered time series were less affected. For consistency, the flux errors should be calculated

based on the same time series (in terms of filtering) as used for flux calculation. Apart from the impact of the low-frequency contribution to flux errors (and fluxes), which we believe is related to non-stationarity of the conditions, we observed that, in order to obtain $\hat{\delta}_{F,FS}$ with good accuracy, integration of Eq. (6) over 200 s is sufficient for wide range of sites as well as observation conditions. Finkelstein and Sims (2001) originally performed summation over 20 s and suggested that the results changed less than 1 to 2% for summation over 10 to 40 s for the dataset they used. Our results suggested that longer

summation period is needed for robust determination of the error in case of tower based measurements over variety of surfaces and wide range of observation conditions.

The EC fluxes are uncertain due to stochastic nature of turbulence by about 10 to 20% under typical observation conditions. By using simulation of time series with statistical properties similar to natural records we deduce that the flux error estimates in turn are uncertain by about 10 to 30%.

**Appendix A. Markovian simulation of time series**

The wind speed and scalar concentration time series were simulated as

$$
\begin{aligned}
w(t) &= \alpha_w w(t - \Delta t) + \beta_w \varepsilon_w(t) \sigma_w \\
c(t) &= \alpha_c c(t - \Delta t) + \beta_{cw} [\alpha_{wc} \varepsilon_w(t) + \beta_{wc} \varepsilon_c(t)] \sigma_c
\end{aligned}
\tag{A1}
$$

where $\varepsilon_w(t)$ and $\varepsilon_c(t)$ are Gaussian random processes with zero means and unit variances, and to preserve the variances and

covariance, the coefficients were chosen as

$$
\begin{aligned}
\beta_x &= \sqrt{(1 - \alpha_x^2)}, \quad x = w, c, wc \\
\alpha_{wc} &= \frac{\rho_{wc}(1 - \alpha_w \alpha_c)}{\beta_w \beta_c}
\end{aligned}
\tag{A2}
$$





Here $\rho_{wc} = \frac{R_{wc}(0)}{\sigma_w \sigma_c}$ is the cross-correlation between $w$ and $c$. By taking $\alpha_w = \exp\left(\frac{-\Delta t}{\tau_w}\right)$ and $\alpha_c = \exp\left(\frac{-\Delta t}{\tau_c}\right)$, where $\Delta t$ is the

simulation time step (much shorter than the time scales $\tau_w$ and $\tau_c$), the constructed processes have the following exponential

covariance and cross-covariance functions:

$$R_w(\tau) = \sigma_w^2 \exp\left(-\frac{|\tau|}{\tau_w}\right)$$
$$R_c(\tau) = \sigma_c^2 \exp\left(-\frac{|\tau|}{\tau_c}\right)$$
$$R_{wc}(\tau) = \begin{cases} u_* c_* \exp\left(-\frac{|\tau|}{\tau_c}\right), \tau > 0 \\ u_* c_* \exp\left(-\frac{|\tau|}{\tau_w}\right), \tau < 0 \end{cases} \qquad \text{(A3)}$$

The theoretical random error estimate for the flux calculated from described artificial time series is given according to Eq.

(5) by

$$\delta_F^2 = 2(\sigma_w^2 \sigma_c^2 + u_* c_*) \left[ \frac{\tau_w \tau_c}{T(\tau_w + \tau_c)} \right], \qquad \text{(A4)}$$

where $u_*$ and $c_*$ represent the friction velocity and the flux concentration, defining the flux by $F = u_* c_*$. For the error

estimate by Wienhold et al. (1995), Eq. (10), the respective error would read as

$$\delta_{F,W}^2 = 2\sigma_w^2 \sigma_c^2 \left[ \frac{\tau_w \tau_c}{T(\tau_w + \tau_c)} \right]. \qquad \text{(A5)}$$

For unstable stratification ($L$<0) the following scaling of variances was assumed (Monin and Yaglom, 1971; Rannik, 1998)

$$\frac{\sigma_w}{u_*} = 1.25 \left(1 - 3\frac{Z}{L}\right)^{1/3}$$
$$\frac{\sigma_c}{c_*} = 3 \left(1 - 28\frac{Z}{L}\right)^{-1/3} \qquad \text{.} \qquad \text{(A6)}$$

Under stable stratification the neutral limits of the above expressions were used.

In addition, the time scales were related to wind speed and stability via

$$n_w = \begin{cases} 0.5, & L < 0 \\ 0.5 + 0.755\zeta, & L > 0 \end{cases}, \qquad \text{(A7)}$$

where the normalised frequency $n = \frac{fZ}{U}$ is used and $\zeta = Z/L$, and

$$n_c = \begin{cases} 0.062, & L < 0 \\ 0.062 + 0.415\zeta^{0.6}, & L > 0 \end{cases}. \qquad \text{(A8)}$$

Time scales and frequencies are related via $f_x = \frac{1}{2\pi\tau_x}$.

**Acknowledgements**

The study was supported by EU projects InGOS and GHG-LAKE (project no. 612642), Nordic Centre of Excellence

DEFROST, and National Centre of Excellence (272041), ICOS (271878), ICOS-FINLAND (281255), ICOS-ERIC

(281250), CarLAC (281196) funded by Academy of Finland. This work was also supported by institutional research funding

(IUT20-11) of the Estonian Ministry of Education and Research. O. Peltola is grateful for Vilho, Yrjö and Kalle Väisälä

foundation for funding.



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







**Table 1**. Flux error estimates $\delta_F$, $\delta_{F,W}$, and $\delta_{F,B}$ obtained from simulations. Height to wind speed ratios $Z\,U^{-1} = 10$ s. Number of simulated realisations N = 10 000. $\tau_w$ and $\tau_c$ denote the time scales of simulated series $w$ and $c$ in Eq. (A1). The values in the parenthesis indicate the uncertainty of the error estimates (obtained from Eq. 16) relative to the average values in percent i.e. $100\% * \sigma(\hat\delta_F)\,\langle\hat\delta_F\rangle^{-1}$.

| $\dfrac{Z}{L}$ | $\tau_w$, $\tau_c$ | 100 x $\delta_F$ $\lvert F\rvert^{-1}$ | | | 100 x $\delta_{F,W}$ $\lvert F\rvert^{-1}$ | | | | 100 x $\delta_{F,B}$ $\lvert F\rvert^{-1}$ | | |
|---|---|---|---|---|---|---|---|---|---|---|---|
| | | Anal. Eq. (A4) | Simul. $\sigma_F$ Eq. (15) | Eq. (6) $\langle\hat\delta_{F,FS}\rangle$ $\pm\,\sigma(\hat\delta_{F,FS})$ | Anal. Eq. (A5) | Simul.[1] $\sigma_F$ Eq. (15) | Eq. (9) $\langle\hat\delta_{F,W}\rangle$ $\pm\,\sigma(\hat\delta_{F,W})$ | Eq. (9)[2] $\langle\hat\delta_{F,W}\rangle$ $\pm\,\sigma(\hat\delta_{F,W})$ | Anal. Eq. (12) | Simul.[3] $\sigma_F$ Eq. (15) | Eq. (11) $\langle\hat\delta_{F,B}\rangle$ $\pm\,\sigma(\hat\delta_{F,B})$ |
| -1 | 0.32, 2.57 | 12.2 | 12.2 | 13.1± 2.9 (22%) | 10.9 | 10.8 | 8.3± 2.5 (30%) | 9.9± 2.6 (26%) | 1.44 | 1.46 | 1.40± 0.27 (19%) |
| 0 | 0.32, 2.57 | 21.8 | 21.7 | 22.2± 4.0 (18%) | 21.0 | 21.0 | 15.6± 4.2 (27%) | 19.5± 4.2 (22%) | 2.80 | 2.81 | 2.71± 0.51 (19%) |
| 1 | 0.13, 0.33 | 12.4 | 12.4 | 12.6± 0.9 (7%) | 12.0 | 11.7 | 11.6± 1.5 (13%) | 12.6± 1.6 (12%) | 2.80 | 2.83 | 2.74± 0.44 (16%) |

[1]Assuming auto-correlated but independent variables $w(t)$ and $c(t)$, with $\alpha_{wc} = 0$.

[2]The method was modified such that in Eq. (9) the co-variances $R_{ws}(t')$ were calculated with 10 s intervals within the lag ranges from -300 to -100 and +100 to +300 s making essentially the estimates independent.

[3]Assuming independent random variables $w(t)$ and $c(t)$, with $\alpha_w = 0$, $\alpha_c = 0$ and $\alpha_{wc} = 0$.





**Table 2.** Linear fitting parameters between absolute value of the flux and flux uncertainty estimates (uncertainty = |flux| x S + I) obtained from measurements at three sites. Robust fitting method was used to minimise the effect of outliers.

| | | CO$_2$ | | | LE | | | CH$_4$ | | |
|---|---|---|---|---|---|---|---|---|---|---|
| | | S (-) | I (μmol m$^{-2}$ s$^{-1}$) | r$^2$ | S (-) | I (W m$^{-2}$) | r$^2$ | S (-) | I (nmol m$^{-2}$ s$^{-1}$) | r$^2$ |
| Forest | $\hat{\delta}_{F,FS}$ | 0.139 | 0.23 | 0.83 | 0.18 | 1.952 | 0.91 | | | |
| | $\hat{\delta}_{F,W}$ | 0.109 | 0.16 | 0.87 | 0.13 | 1.332 | 0.93 | | | |
| | $\hat{\delta}_{F,B}$ | 0.013 | 0.06 | 0.66 | 0.02 | 0.358 | 0.85 | | | |
| | $\hat{\delta}_{F,N}$ | 0.004 | 0.03 | 0.35 | 0.01 | 0.070 | 0.45 | | | |
| Fen | $\hat{\delta}_{F,FS}$ | 0.084 | 0.15 | 0.75 | 0.09 | 1.767 | 0.91 | 0.112 | -0.55 | 0.99 |
| | $\hat{\delta}_{F,W}$ | 0.076 | 0.08 | 0.82 | 0.08 | 1.093 | 0.94 | 0.092 | -0.28 | 0.98 |
| | $\hat{\delta}_{F,B}$ | 0.015 | 0.03 | 0.64 | 0.02 | 0.209 | 0.91 | 0.017 | 0.10 | 0.98 |
| | $\hat{\delta}_{F,N}$ | 0.005 | 0.01 | 0.30 | 0.01 | 0.022 | 0.52 | 0.002 | 0.20 | 0.02 |
| Lake | $\hat{\delta}_{F,FS}$ | 0.343 | 0.14 | 0.75 | 0.09 | 1.827 | 0.76 | | | |
| | $\hat{\delta}_{F,W}$ | 0.206 | 0.10 | 0.72 | 0.09 | 1.196 | 0.75 | | | |
| | $\hat{\delta}_{F,B}$ | 0.030 | 0.04 | 0.55 | 0.02 | 0.078 | 0.79 | | | |
| | $\hat{\delta}_{F,N}$ | | | | | | | | | |





**Table 3.** Amount and magnitude of reliable noise estimates obtained by using Lenschow et al. (2000) method. The reported values are estimates of 1 std at 10 Hz sampling rate (i.e. $\hat{\sigma}_{n,f=10Hz}$). Noise estimate was considered reliable if SNR<3, where SNR was calculated using the modal value of $\hat{\sigma}_{n,f=10Hz}$ obtained from the measurements. Also the performance specifications reported by instrument manufacturers are given for comparison. They are RMS values of 10 Hz data if not otherwise noted.

| | | % of reliable estimates (%) | Median value | 25th percentile | 75th percentile | Specifications of analysers |
|---|---|---|---|---|---|---|
| Forest | u (m s$^{-1}$) | 0 | | | | |
| | T$_s$ (°C) | 3 | 0.023 | 0.021 | 0.025 | |
| | CO$_2$ (ppm) | 61 | 0.28 | 0.27 | 0.28 | 0.6[1] |
| | H$_2$O (ppth) | 11 | 0.015 | 0.015 | 0.016 | 0.06[1] |
| Fen | u (m s$^{-1}$) | 6 | 0.062 | 0.052 | 0.066 | |
| | T$_s$ (°C) | 15 | 0.048 | 0.041 | 0.059 | |
| | CO$_2$ (ppm) | 57 | 0.23 | 0.23 | 0.24 | 0.16 |
| | H$_2$O (ppth) | 1 | 0.011 | 0.010 | 0.011 | 0.011 |
| | CH$_4$ (ppb) | 5 | 1.67 | 1.60 | 1.78 | 9.5[2] |
| Lake | u (m s$^{-1}$) | 0 | | | | |
| | T$_s$ (°C) | 3 | 0.034 | 0.030 | 0.038 | |
| | CO$_2$ (ppm) | 0 | | | | 0.11 |
| | H$_2$O (ppth) | 0 | | | | 0.0047 |

[1] Peak-to-peak value. For the comparison with 1 std precision characteristic should be divided by a factor of 3.

[2] Converted from a value reported for 1 Hz data using Eq. (8)





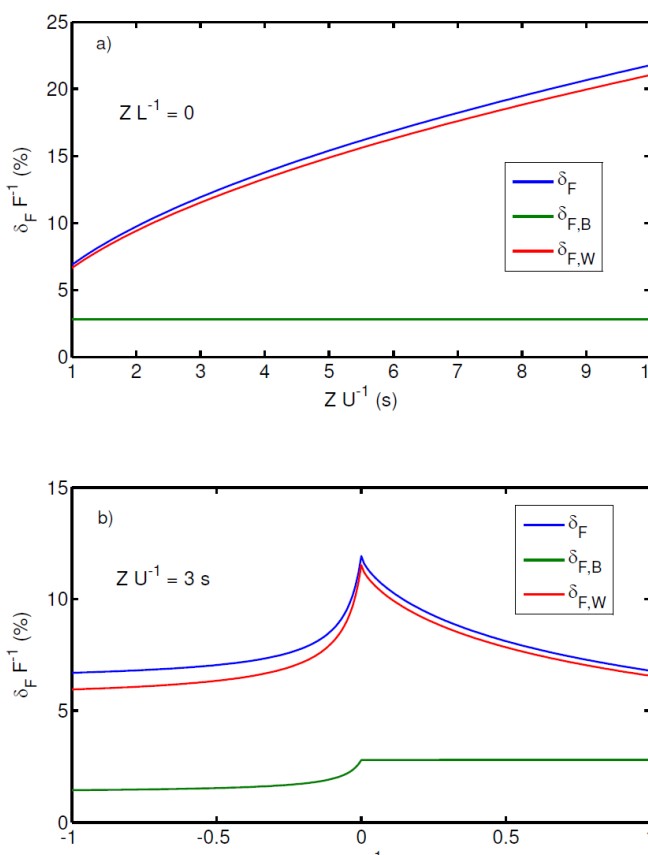

**Figure 1.** Dependence of relative flux random errors on a) $Z\,U^{-1}$ and b) $Z\,L^{-1}$. The error estimates $\delta_F$, $\delta_{F,B}$ and $\delta_{F,W}$ were calculated according to Eqs. (A4), (12) and (A5), respectively.



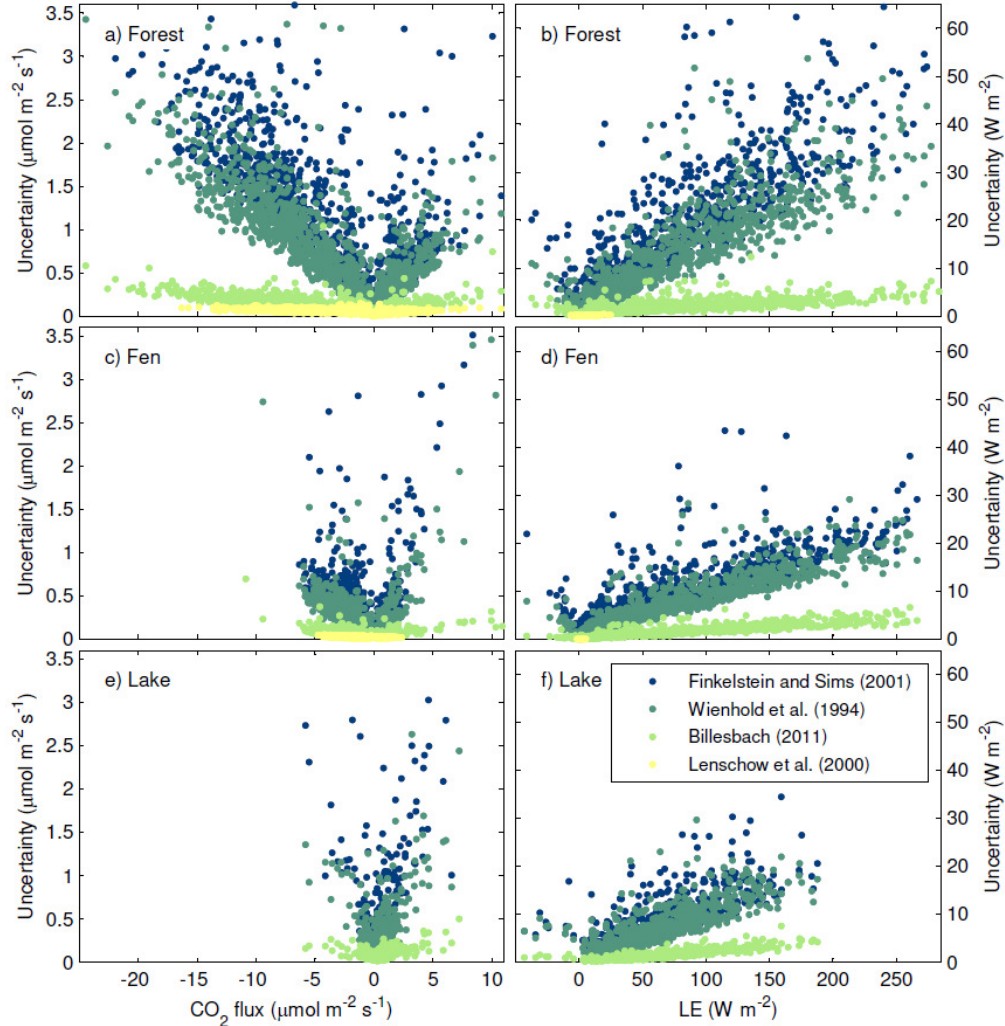

**Figure 2.** Flux error estimates $\hat{\delta}_{F,FS}$, $\hat{\delta}_{F,W}$, $\hat{\delta}_{F,B}$ and $\hat{\delta}_{F,N}$ versus the flux magnitude. Individual outliers were left outside the plots in order to show the majority of the data better.





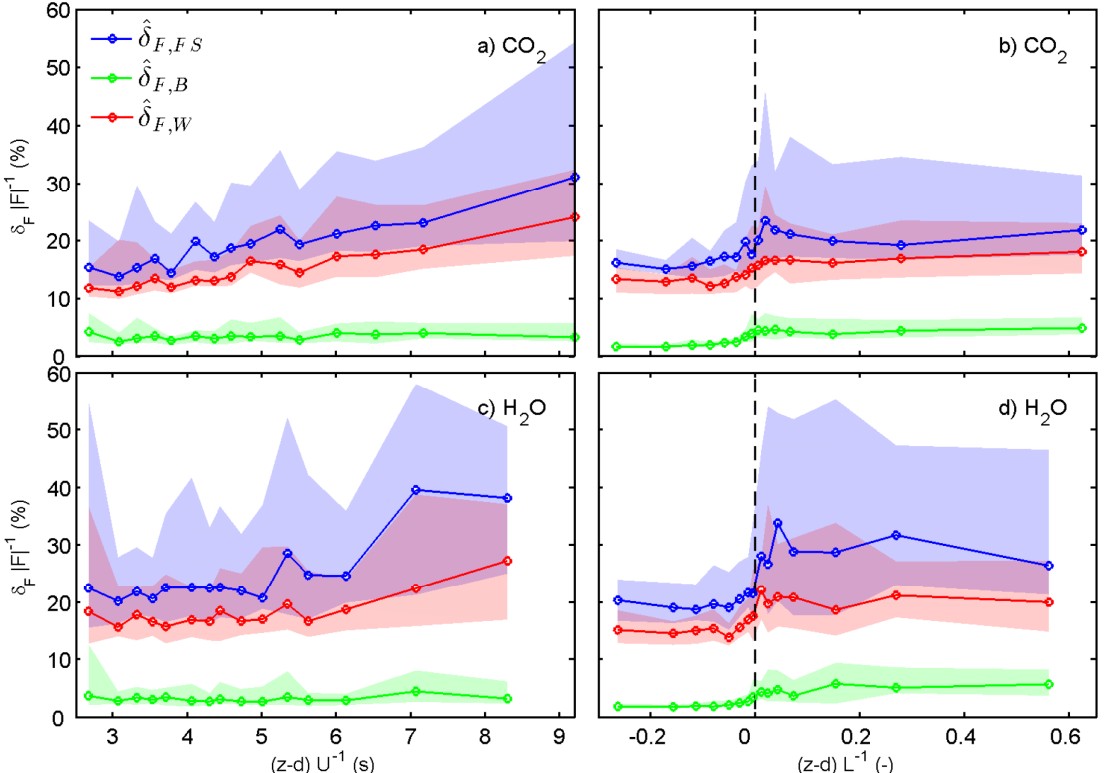

**Figure 3.** Dependence of relative errors on *(z-d) U^{-1}* and *(z-d) L^{-1}* at the SMEAR II forest site. Data were binned before plotting (circles: medians, areas: interquartile range). For subplots a) and c) only periods with $|(z-d) L^{-1}|<0.1$ were used, and for subplots b) and d) periods when $2 \text{ m s}^{-1} < U < 4 \text{ m s}^{-1}$ were selected.





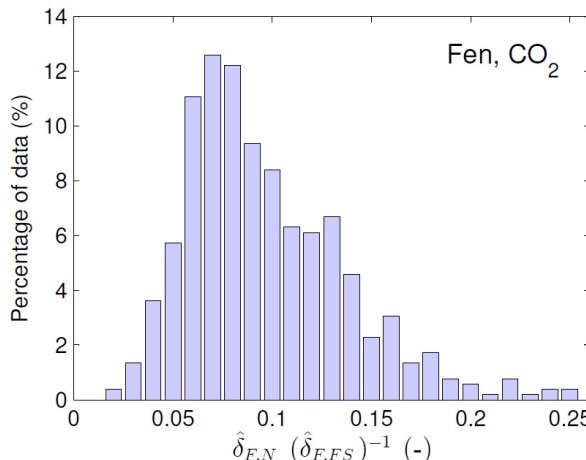

**Figure 4.** Frequency distribution of the ratio between the flux instrumental noise error (the error estimate $\hat{\delta}_{F,N}$) and total

flux uncertainty (the error estimate $\hat{\delta}_{F,FS}$). $CO_2$ data measured at the fen site with SNR<3 (see Sect. 4.3.1) were used in the

5     plot.





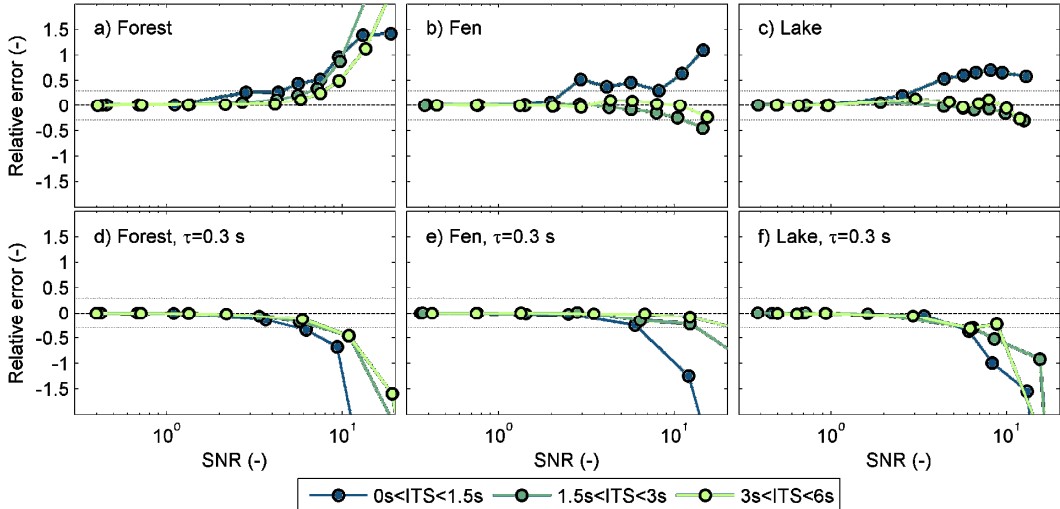

**Figure 5.** Relative difference between the estimated ($\hat{\sigma}^2_{n,f}$) and the actual ($\sigma^2_{n,f}$) signal noise variance as a function of SNR and ITS calculated as $\frac{z-d}{2\pi n_m U}$, where $n_m$ was found according to Eq. (3). The data were bin averaged before plotting.

5    The values for $\tau$ in the subplots show the time constant used in low-pass (auto-regressive) filtering of the original temperature time series (i.e. before superimposing the noise). The dashed lines highlight the zero line (no systematic error in the noise estimate), whereas the dotted lines show the ±30 % thresholds. The top row shows data with no low-pass filtering.



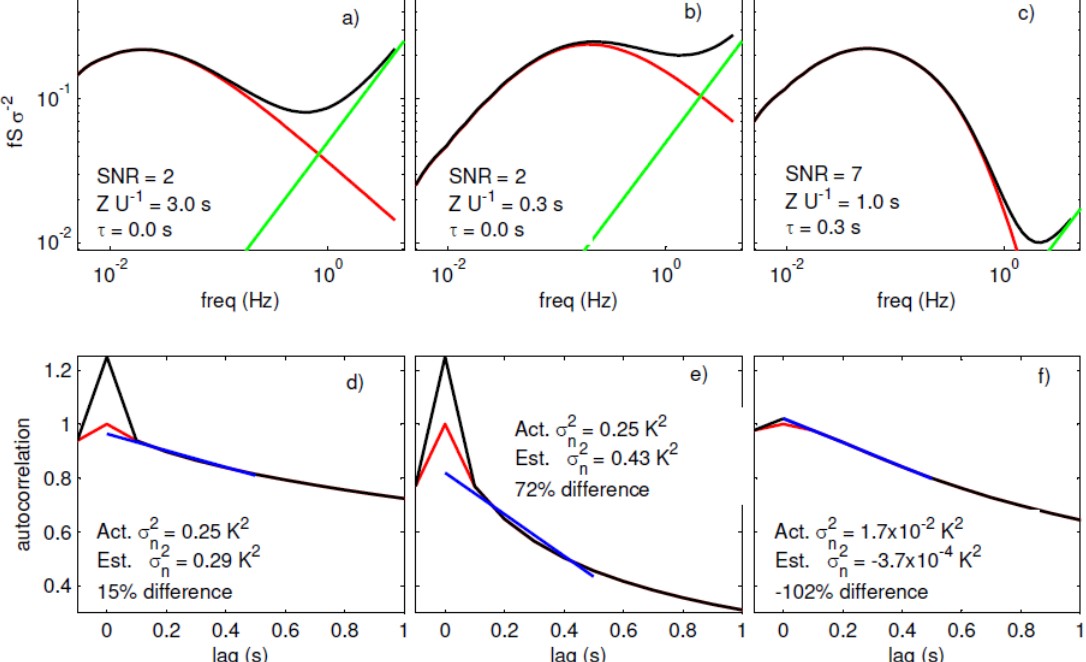

**Figure 6.** Three example cases for application of the Lenschow et al. (2000) method with presented spectra (upper plots) and auto-correlation functions (lower plots). Case 1 (plots a, d): No filtering, low SNR, high ITS. Case 2 (plots b, e): No filtering, low SNR, low ITS. Case 3 (plots c, e): Filtering used, high SNR, moderate ITS. Red lines correspond to the original signal; Black lines correspond the original signal (high-passed filtered in case 3) superimposed with the noise; Blue lines: linear fits to the auto-correlation functions for lags from 0.1 to 0.5 s; Green line in spectral plots indicates the +1 slope on log-log representation.





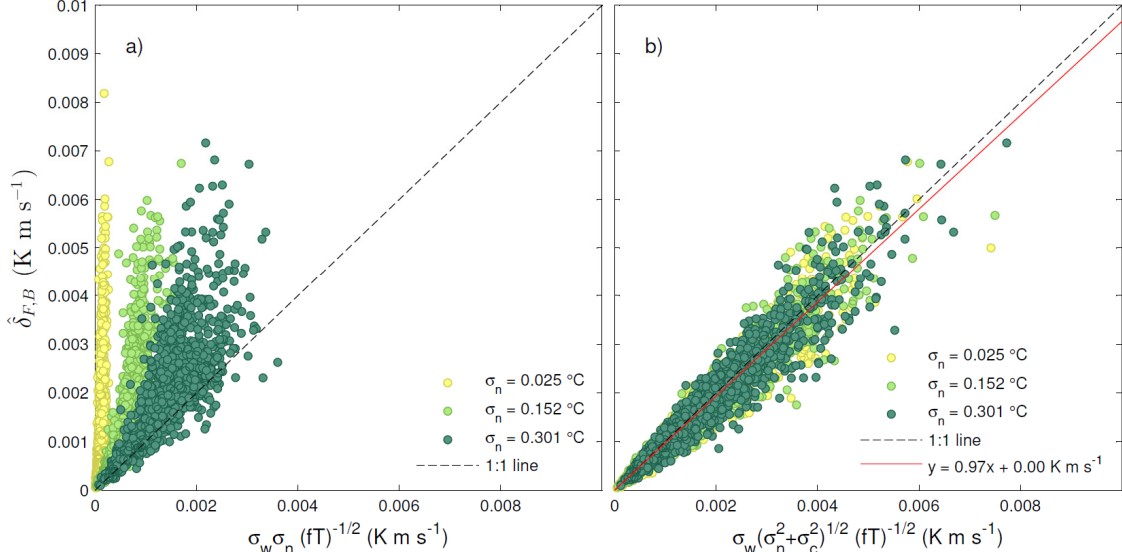

**Figure 7.** Error estimate calculated based on the Billesbach (2011) method ($\hat{\delta}_{F,B}$, Eq. (11) in this study) plotted against flux uncertainty caused by instrumental noise (subplot a) and a combination of noise and signal variances (subplot b). If the method would estimate purely the instrumental noise, the points in the subplot a) would follow 1:1 line. Instead, as predicted by Eq. (12), the points follow 1:1 line in the subplot b). Sonic temperature data from the Forest site was used with different level of noise added as shown in the legend.



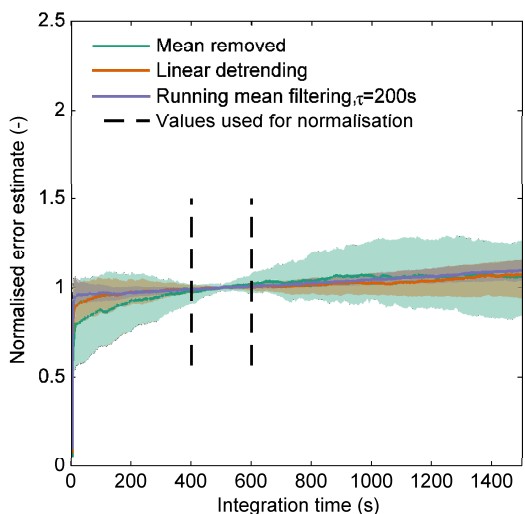

**Figure 8.** Normalised errors $\hat{\delta}_{F,FS}$ up to integration time 1500 s. Estimated from time series with 2 hrs (7 200 s) duration to avoid large uncertainties of the auto-covariance function at long lags. Normalisation is done with the average between 400 and 600 s. Lines show medians and the areas interquartile ranges around the medians. The time constant used in the high pass filtered case is given in the plot. Sonic temperature data from the Forest site were used.

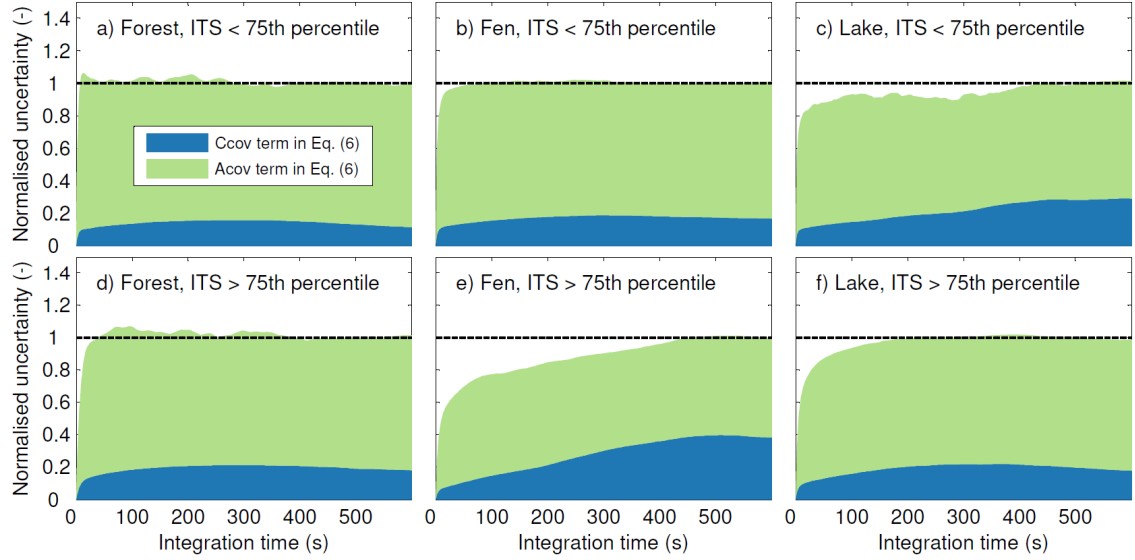

**Figure 9.** Normalised uncertainty estimate according to Finkelstein and Sims (2001), $\hat{\delta}_{F,FS}$, as a function of integration time used in Eq. (6). Contribution of the cross-covariance (blue area) and auto-covariance (green area) terms in the Eq. (6)



are shown separately. The values were normalised with average $\hat{\delta}_{F,FS}$ at integration times between 400…600 s and grouped

in two ITS classes before plotting. No detrending of time series was used.