# Peer review of "Random uncertainties of flux measurements by the eddy covariance technique"

_Atmospheric Measurement Techniques, 2016_

## Referee Comment (RC1) · Anonymous Referee #1 · 16 Mar 2016

This manuscript describes the authors research into several different methods of determining various components of the uncertainty in eddy covariance fluxes. Until recently, this has been a badly neglected subject and it is good to see interest from the community.

While this is overall a very good paper, I find that there are a few issues that could use further explanation and clarification. One issue is contained in the authors equation 8. Through this, they imply that arbitrarily small instrument noise can be obtained through longer and longer integration times. While this is true for a limited range of integrations, it is not true in general. In fact, this is the basis of Allen Variance plots which show the actual noise floor for an instrument. This should be made clear in the text, and they should offer some evidence that their computations do not violate the limits where eqn. 8 holds.

Another issue is their statement that "...the noise component of the vertical wind speed measurement is negligible." While this is small, it must be viewed in the context of usually small vertical wind speed fluctuations. If this were true, then the same could be said of horizontal wind speeds and sonic temperatures which all derive from the same fundamental measurement (sound pulse transit times). If they wish to stand by this statement, they need to provide evidence that it is true, especially in the context of other sonic anemometer parameters. One problem that may contribute to this confusion is the nature by which sonic anemometer data are often recorded. If the wind speeds (and temperature or speed of sound) are recorded digitally, the data streams often are comprised of ASCII character strings where the data are truncated. This may give the impression that the instrument has little or no noise for a particular measurement, but examination of the same data stream, formatted as a binary output might show otherwise. How would a significant noise factor in vertical wind speed change equations 7 and 8?

When the authors discuss the "shuffle" method, they claim that equation 11 is equivalent to equation 7. Where is the justification for this. It's not clear that this is so, and a better derivation would be helpful here. Finally it seems that equations 7 and 12 assume perfect correlation between the noise components of vertical wind speed and "s". One often seen definition of correlation coefficient is the ratio of equation 11 to equation 7. Why does this factor disappear in this analysis? In section 4.3.2, the authors assert that the "shuffle" method over-estimates the instrument system noise because it includes residual turbulent fluctuation information. This would be expected from the description contained in Billesbach's paper. In it, they show that some level of averaging (over different ensembles) is needed to generate a robust noise estimate. The question then arises "What level of averaging did the current authors use, and would they arrive at different conclusions if they included a larger ensemble sample in their analysis?"

These are all issues that ought to be considered by the authors. They are questions

that will arise for many readers, and addressing them will certainly add to the useful-ness of this already nice work.

---

## Referee Comment (RC2) · Anonymous Referee #2 · 3 May 2016

General comments

This paper compares different methods for estimating the random error of eddy covariance flux estimates. Although the paper is well written, the novelty of the presented research remains unclear. I cannot find substantial new insights compared to the study of Billesbach (2011). Indeed, the Wienhold et al. (1995) has also been compared, but only to reach at the conclusion that it gives almost the same error estimate as the well-established method of Finkelstein and Sims (2001). With respect to the error for instrumental noise, it was already proposed by Mauder et al. (2013) to apply the approach of Lenschow et al. (2000) for eddy-covariance measurements and not only for lidar data, for which it had been developed. So, this aspect is also not really innovative. Moreover, one other important method for estimating the random flux error has not been considered at all. Salesky et al. (2012) proposed a promising method that is

based on local flux decomposition and demonstrated clearly its advantages over other existing methods, such as the ones of Finkelstein and Sims (2001), because it does not require an estimate of the integral time scale. However, the Salesky paper is not even cited anywhere in the manuscript. In summary, due to the lack of novelty and due to an incomplete consideration of existing literature, I cannot recommend this manuscript for publication in this journal.

Salesky S, Chamecki M, Dias N (2012) Estimating the random error in eddy-covariance based fluxes and other turbulence statistics: the filtering method. Boundary-Layer Meteorol 144:113–135.

---

## Author Comment (AC1) · 14 Jun 2016

We thank the reviewer for the critical assessment of our paper. The reviewer refers to two main limitations of the paper: lack of novelty in results and omitting of one of the alternative methods proposed by Salesky et al. (2012) to estimate the random flux error. Whereas we agree with the last comment (see the answer below), we do not agree with the general statement that the paper has no value. Our main aim was to bring clarity among the methods frequently used for evaluation of random flux errors. We approached this in a systematic way giving theoretical overview of all methods as well as numerical evaluation. In our opinion different flux error estimation methods have been used in literature without clear understanding how different methods relate to each other and what is the error estimate produced by each of the methods. In summary, we thank the reviewer for bringing into our attention important deficiency of

our paper while omitting the method by Salesky et al. (2012). We will amend the paper with respective theory subsection and results. We believe that after such revision the scientific community working with EC flux measurements will benefit from this paper when interpreting the flux data and associated random errors as estimated by different methods.

General comments

Referee Comment (RC): This paper compares different methods for estimating the random error of eddy covariance flux estimates. Although the paper is well written, the novelty of the presented research remains unclear. I cannot find substantial new insights compared to the study of Billesbach (2011). Indeed, the Wienhold et al. (1995) has also been compared, but only to reach at the conclusion that it gives almost the same error estimate as the well-established method of Finkelstein and Sims (2001).

Author Response (AR): Yes, we agree that the paper by Billesbach (2011) compares different flux error evaluation methods. Billesbach proposes also the method that, according to the author, measures the instrument uncertainty. In the current manuscript we evaluate the method in comparison to the method first proposed by Lenschow et al. (2000) and demonstrate both theoretically and empirically by using measured data that it produces a different error estimate.

RC: With respect to the error for instrumental noise, it was already proposed by Mauder et al. (2013) to apply the approach of Lenschow et al. (2000) for eddy-covariance measurements and not only for lidar data, for which it had been developed. So, this aspect is also not really innovative.

AR: The authors refer to Mauder et al. (2013) in connection to the method proposed by Lenschow et al. (2000). The contribution of our manuscript to the method by Lenschow et al. (2000) and Mauder et al. (2013) is to evaluate and present the applicability range of the method which clearly depends on SNR.

RC: Moreover, one other important method for estimating the random flux error has not been considered at all. Salesky et al. (2012) proposed a promising method that is based on local flux decomposition and demonstrated clearly its advantages over other existing methods, such as the ones of Finkelstein and Sims (2001), because it does not require an estimate of the integral time scale. However, the Salesky paper is not even cited anywhere in the manuscript. In summary, due to the lack of novelty and due to an incomplete consideration of existing literature, I cannot recommend this manuscript for publication in this journal.

AR: The method by Salesky et al. (2012) has been unfortunately out of our attention. After careful examination of the method, we recognise that it is certainly a good alternative to the other methods. In addition, it does not require estimation of the integral time scale and has an advantage compared to the methods which do so. As a comment, the method proposed by Lenschow and Kristensen (1985) and in a discrete form by Finkelstein and Sims (2001) does not require either direct evaluation of the time scale but just integration of the products of the co-variance functions over sufficiently long time exceeding the integral time scale. The method by Salesky et al. (2012) is a novel approach by introducing filtering method in error evaluation. However, the method stems from the same theory and is therefore expected to result in the same error estimates as Finkelstein and Sims (2001) under stationary conditions.

While avoiding direct calculation of the integral time scale, it is indirectly embedded in the method by Finkelstein and Sims (2001) and the estimation result is also affected by the uncertainty that the non-stationarity introduces. The same applies to the method by Salesky et al. (2012) because the instantaneous turbulent flux w'c'(t) is also affected by non-stationarity. It was demonstrated by Salesky et al. (2012) that under non-stationery conditions their method performed better than the other methods involving direct evaluation of the integral time scale, namely the methods by Lumley and Panofsky (1964) and Lenschow et al. (1994).

We believe the method proposed by Salesky et al. (2012) is by no means a very

promising method and a reliable alternative for flux error evaluation. However, its superior performance is not directly evident compared to the method by Finkelstein and Sims (2001). The theory of turbulent flux and flux error calculation relies on the assumption of the stationarity. Non-stationary conditions introduce additional uncertainty in time-averages. In the revised MS we present the results also for the method by Salesky et al. (2012) and compare with the results produced by the method by Finkelstein and Sims (2001) to see if and under what conditions the methods deviate.

Lenschow, D., Mann, J., Kristensen, L.: How long is long enough when measuring fluxes and other turbulence statistics? J Atmos Ocean Technol 11(3):661–673, 1994.

Lumley, J., Panofsky, H., The structure of atmospheric turbulence. Interscience, New York, 239 pp, 1964.

Salesky, S., Chamecki, M., Dias, N., Estimating the random error in eddy-covariance based fluxes and other turbulence statistics: the filtering method. Boundary-Layer Meteorol 144:113–135, 2012.

---

## Author Comment (AC2) · 14 Jun 2016

1. While this is overall a very good paper, I find that there are a few issues that could use further explanation and clarification. One issue is contained in the authors equation 8. Through this, they imply that arbitrarily small instrument noise can be obtained through longer and longer integration times. While this is true for a limited range of integrations, it is not true in general. In fact, this is the basis of Allen Variance plots which show the actual noise floor for an instrument. This should be made clear in the text, and they should offer some evidence that their computations do not violate the limits where eqn. 8 holds.

Equation (8) applies under the assumption that the instrumental noise is the white noise. The white noise has been frequently observed at power spectra as the +1 slope when the spectrum is weighed with the frequency. In natural signals turbulent variation and noise are combined in the signal and therefore the power spectra as well as Allan variance plots show the characteristics of the combined signal. Therefore eq. (8) cannot be utilised for natural records. However, for theoretical consideration of the white noise the equation holds because by definition the white noise has equal power at all frequencies.

We clarify this in the manuscript stating that the eq. (8) is valid for white noise.

2. Another issue is their statement that "...the noise component of the vertical wind speed measurement is negligible." While this is small, it must be viewed in the context of usually small vertical wind speed fluctuations. If this were true, then the same could be said of horizontal wind speeds and sonic temperatures which all derive from the same fundamental measurement (sound pulse transit times). If they wish to stand by this statement, they need to provide evidence that it is true, especially in the context of other sonic anemometer parameters.

The noise level of the sonic anemometers is typically a few hundredths of m s$^{-1}$. For example, Rannik et. al. (2015) reported the noise level of an anemometer USA1 by METEK to be 0.037 m s$^{-1}$ at 10 Hz sampling frequency for the vertical wind speed component and a similar value for the anemometer model R3-50, Gill Instruments, Ltd., Hampshire, UK.

In analogy to eq. (7) in the manuscript, the flux error due to instrumental noise can be written as

$$\delta_{F,N} = \frac{\sqrt{\sigma^2_{n\_w,f}\sigma^2_c + \sigma^2_w\sigma^2_{n\_c,f}}}{\sqrt{fT}}$$ , where subscripts $n\_w$ and $n\_c$ represent the instrumental noise of

the vertical wind speed and scalar concentration measurements. Typical values of $\sigma_w$ vary between 0.1 to 1 m s$^{-1}$. This yields for the sonic anemometers the signal-to-noise ratio values as defined in the MS,

$$SNR_w = \frac{\sigma_w}{\sigma_{n\_w,f}}$$ , in the order from 1 (if to assume very small $\sigma_w$ 0.05 m s$^{-1}$ and $\sigma_{n\_w,f}$ 0.04 m s$^{-1}$ at

10 Hz) to 25. Following the atmospheric similarity relationships (under near neutral conditions) $\sigma_w = 1.25u_*$ and $\sigma_c = 3c_*$, the expression for relative flux error can be written as

$$\frac{\delta_{F,N}}{|F|} = \frac{1.25 \times 3\sqrt{\left(SNR^2_w\right)^{-1} + \left(SNR^2_c\right)^{-1}}}{\sqrt{fT}}$$ , where SNR with subscripts $w$ and $c$ denote the values for

wind speed and scalar concentration measurements. It can be seen that the relative contribution of the anemometer and gas analyser noise depends on the respective SNR values and the higher the SNR the smaller is the contribution to the flux error. The estimated SNR values for the sonic anemometers are typically larger than the respective values for the gas analysers under most of the observation conditions and the relative flux error resulting from the sonic anemometer noise is thus estimated to be in the order of 0.1 to 2% for 30 min averaging period. This is small enough to be negligible from the practical point of view. The same applies for the noise estimated for the sonic temperature measurements (Table 3 in the manuscript) and can be the case also with many other scalars. However, we meant that the noise of the modern sonic anemometers does not contribute significantly to the flux error and can be usually ignored.

We modify the eq. (7) in the manuscript to include also the effect of the anemometer's noise as given above.

One problem that may contribute to this confusion is the nature by which sonic anemometer data are often recorded. If the wind speeds (and temperature or speed of sound) are recorded digitally, the data streams often are comprised of ASCII character strings where the data are truncated. This may give the impression that the instrument has little or no noise for a particular measurement, but examination of the same data stream, formatted as a binary output might show otherwise. How would a significant noise factor in vertical wind speed change equations 7 and 8?

We modify the equation 7 to include also the contribution of the anemometer noise in flux error, see the previous answer.

3. When the authors discuss the "shuffle" method, they claim that equation 11 is equivalent to equation 7. Where is the justification for this. It's not clear that this is so, and a better derivation would be helpful here.

The equation (11) is a method proposed by Billesbach (2011) and included random re-ordering ("shuffling") of one variable with respect to each other, which makes the two time series fully uncorrelated.

Let us assume discrete time series $w$ and $s$ (which have zero means for simplicity), which have variances $\sigma_w$ and $\sigma_s$. After random shuffling the series $s$ it will have the same variance as before shuffling but it is uncorrelated in time (the correlation function is 0 except at zero lag equals to 1). Random shuffling makes also $s$ uncorrelated with respect to $w$. By definition the variance of the product of two independent variables (with zero means) is the product of the variances of the variables, i.e. if $\varphi = w\,s$, then $\sigma_\varphi^2 = \sigma_w^2 \sigma_s^2$. The error of the average of $\varphi$ (the standard error) over $N$ realisations is given by $\delta_\varphi = \dfrac{\sigma_\varphi}{\sqrt{N}}$, which, assuming the time series with length $T$ sampled at frequency $f$ ( $N = fT$ ), gives the error estimate of time-average $\varphi$ as $\delta_\varphi = \dfrac{\sigma_w \sigma_s}{\sqrt{fT}}$. Thus eq. (11) becomes equivalent to (7) when replacing $n$ with $s$ in eq. (7).

Finally it seems that equations 7 and 12 assume perfect correlation between the noise components of vertical wind speed and "s". One often seen definition of correlation coefficient is the ratio of equation 11 to equation 7. Why does this factor disappear in this analysis?

Equation 7 and 12 assume that the noise of the vertical wind component is negligible (thus taken zero). Thus it does not assume perfect correlation of the noise components of $w$ and $s$. For the error estimate including the noise in vertical wind speed see the answer to the comment 2 above.

In section 4.3.2, the authors assert that the "shuffle" method over-estimates the instrument system noise because it includes residual turbulent fluctuation information. This would be expected from the description contained in Billesbach's paper. In it, they show that some level of averaging (over different ensembles) is needed to generate a robust noise estimate. The question then arises "What level of averaging did the current authors use, and would they arrive at different conclusions if they included a larger ensemble sample in their analysis?"

The random shuffle method essentially treats the turbulent variation as noise and thus the method does not produce equivalent error estimate to the method by Lenschow et al. (2000) and Mauder et al. (2013). The method by Lenschow et al. (2000) gives the uncertainty of the covariance due to instrumental noise

under the hypothetical conditions of no turbulent fluctuations and has thus (to our opinion) clear physical meaning. We were not able to give clear interpretation to the error estimate by Billesbach (2011).

We used 20 repetitions when calculating the uncertainty estimates with random shuffle method, which should be enough to obtain robust estimates (Billesbach, 2011). However, the amount of repetitions only decreases the uncertainty of the error estimate and it does not change the fact that the method treats turbulent variation as noise and thus overestimates the flux error related to instrumental noise only.

Rannik, Ü., Haapanala, S., Shurpali, N., Mammarella, I., Lind, S., Hyvönen, N., Peltola, O., Zahniser, M., Martikainen, P. J., and Vesala, T.: Intercomparison of fast response commercial gas analysers for nitrous oxide flux measurements under field conditions, Biogeosciences. 12, 415-432, 2015.